# Distinct Tumor-Associated Macrophage Signatures Shape the Immune Microenvironment and Patient Prognosis in Renal Cell Carcinoma

**DOI:** 10.3390/cells14211740

**Published:** 2025-11-06

**Authors:** Youngsoo Han, Aidan Shen, Cheng-chi Chao, Lucas Yeung, Aliesha Garrett, Jianming Zeng, Satoru Kawakita, Jesse Wang, Zhaohui Wang, Alireza Hassani, Xiling Shen, Chongming Jiang

**Affiliations:** 1Terasaki Institute for Biomedical Innovation, Los Angeles, CA 90024, USA; youngsoo.han@terasakicolab.org (Y.H.); aidan.shen@terasakicolab.org (A.S.); great_scientist@yahoo.com (C.-c.C.); lucascyyeung@gmail.com (L.Y.); aoraw@terasaki.org (A.G.); skawakita@terasaki.org (S.K.); woopeejesse@gmail.com (J.W.); zhaohui.wang@terasaki.org (Z.W.); hassania@terasaki.org (A.H.); 2Bioengineering Department, University of California, Los Angeles, CA 90024, USA; 3GI Medical Oncology, The University of Texas MD Anderson Cancer Center, Houston, TX 77030, USA; jzeng3@mdanderson.org

**Keywords:** tumor-associated macrophages, renal cell carcinoma, prognosis, tumor immune microenvironment, machine learning

## Abstract

Renal cell carcinoma (RCC) accounts for 90% of adult renal cancer cases and is characterized by significant heterogeneity within its tumor microenvironment. This study tests the hypothesis that tumor-associated macrophages (TAMs) influence RCC progression and patient response to treatment by investigating the prognostic implications of TAM signatures. Utilizing independent single-cell RNA sequencing data from RCC patients, we developed eight distinct TAM signatures reflective of TAM presence. A LASSO Cox regression model was constructed to predict survival outcomes, evaluated using the TCGA dataset, and validated across independent RCC cohorts. Model performance was assessed through Kaplan–Meier survival plots, receiver operating characteristic (ROC) curves, and principal component analysis. Survival analysis demonstrated that specific TAM signature gene expressions serve as significant prognostic markers, identifying TAM signatures positively correlated with patient survival and macrophage infiltration. A 27-gene TAM risk model was established, successfully stratifying patients into risk categories, with low-risk patients showing improved overall survival. These findings provide insights into the role of TAMs in modulating the RCC tumor immune microenvironment and their impact on patient prognosis, suggesting that TAM-based signatures may serve as useful prognostic markers and potential targets to enhance RCC treatment strategies.

## 1. Introduction

Tumor-associated macrophages (TAMs) are a critical component of the tumor immune microenvironment (TIME), playing diverse and often conflicting roles in cancer development and progression. TAMs originate from resident tissue-specific macrophages or monocytes recruited to the tumor site, where they adapt to the TIME and exhibit either pro-inflammatory or anti-inflammatory phenotypes [1,2,3]. These phenotypes, commonly referred to as M1 and M2 macrophages, respectively, influence tumor progression by modulating immune responses, promoting angiogenesis, and facilitating immune evasion [4,5,6]. Given their central role in cancer biology, TAMs have emerged as a key focus for understanding tumor behavior and developing novel therapeutic strategies [7,8].

In renal cell carcinoma (RCC), a prevalent form of kidney cancer, TAMs could significantly contribute to tumor progression and patient prognosis [9,10]. TAMs in RCC promote tumor growth and metastasis by enhancing epithelial–mesenchymal transition (EMT) and cell invasion through pathways such as IL-6/STAT3 signaling [11,12]. Additionally, they suppress immune responses by producing cytokines such as IL-10 and CCL2, impairing natural killer (NK) cell function and fostering an immunosuppressive microenvironment [13,14]. High infiltration of certain TAM subsets correlates with poor patient outcomes, while others may associate with improved survival, highlighting the dual roles TAMs play in RCC progression [7,15,16,17,18,19].

Beyond their roles in tumor biology, TAMs offer significant potential as therapeutic targets and biomarkers [9,17]. Strategies aimed at modulating TAM activity include depleting TAMs, inhibiting their recruitment, or reprogramming them from a pro-tumor M2 phenotype to an anti-tumor M1 phenotype, with the goal of reducing tumor growth and enhancing the efficacy of immunotherapies [5,12,20]. Emerging approaches, such as macrophage-derived nanosponges, have shown promise in modulating TAM behavior and enhancing anti-tumor immunity in RCC [21]. Furthermore, the expression profiles of TAM-related genes and their association with immune checkpoint provide opportunities to refine patient stratification and predict responses to immune checkpoint inhibitors (ICIs) [7,8,15,22].

Despite these advances, the mechanisms by how the TAMs contribute to RCC progression remain incompletely understood [8,10,23,24]. This study focuses on elucidating the roles of TAM signatures in RCC, leveraging their expression profiles to stratify patients based on risk scores, and investigating their interactions with immune components within the TME. By integrating computational deconvolution methods and survival analyses, this work aims to provide insights into the prognostic and therapeutic significance of TAMs in RCC. These findings could pave the way for the development of TAM-targeted therapies, offering new avenues for improving outcomes in RCC patients. A 27-gene risk score derived from TAM signatures across independent datasets can effectively classify patients into low- and high-risk categories, distinguish overall survival, and investigate the potential mechanisms underlying the role of TAMs in the prognosis of RCC patients. The workflow of the present study is illustrated in Figure 1.

## 2. Materials and Methods

### 2.1. Data Utilization

Level 3 RNA-seq data and clinical information for The Cancer Genome Atlas Kidney Renal Clear Cell Carcinoma (TCGA-KIRC, tumor samples number = 537) dataset were obtained from TCGA on FireBrowse (gdac.broadinstitute.org/), and their accompanying clinical information is also shown in the Appendix A. Two independent single-cell RNA-seq datasets and clinical information from human RCC samples were obtained from previous publications [25,26]. The gene expression of 101 RCC patients and related clinical information data were collected from the Sato et al. study named E-MTAB-1980 [27]. Macrophage regulation scores, leukocyte and lymphocyte infiltration scores, and IFN-γ response and TGFβ response scores for TCGA-KIRC samples were downloaded as a Appendix A from prior work [28].

### 2.2. Curation of Immune-Related Genes (IRGs)

Immune-related genes (IRGs) were obtained from Charoentong et al. [29], Bindea et al. [30], and Xu et al. [31]. All genes from immune cells were collected (marker genes attributed to cancer cells were excluded) and combined into a single list of 831 IRGs genes.

### 2.3. Immune Cell Inference

Immune cell fractions in tumor samples were quantified using the quanTIseq method [32], an RNA-seq deconvolution algorithm implemented in R. quanTIseq applies constrained least squares regression to estimate the proportions of immune cell populations, including macrophages, neutrophils, dendritic cells, CD8+ T cells, regulatory T cells (Tregs), and other immune subsets [32]. Using patient bulk gene expression data, quanTIseq enabled identification and correlation analysis between TAM signatures and specific immune cell types, providing insights into immune cell interactions within the tumor microenvironment. Immune infiltration scores of six immune cells were calculated using Binding Association with Sorted Expression (BASE) [33,34,35,36,37], a rank-based gene set enrichment method. Previous publications have detailed and validated immune cell infiltration using this method [6,11,12,13,14,33]. BASE infers immune-cell infiltration for each patient by combining immune cell–specific weight profiles with the patient’s gene expression data. For a given profile, genes are ranked from the highest to the lowest expression and then weighted using the corresponding immune-cell weights. BASE computes two cumulative running-sum curves: a foreground curve for the patient’s weighted expression values and a background curve for the complementary weights (1—weight). When infiltration from a specific immune cell is high, the foreground curve rises steeply—because highly expressed genes tend to carry large weights for that cell—whereas the background curve increases more slowly. The immune-infiltration score is the maximal absolute difference between these two curves, which is then normalized to enable comparison across samples and cell types. This procedure yields one score per immune cell type per patient. Full computational and validation details are provided in the original BASE description and in [11,38]. Similarly, BASE was used to calculate single cell-based TAM scores using TAM signatures. The identical parameter settings across cohorts were applied.

### 2.4. Generation of TAM Signatures

RCC single-cell RNA-seq datasets from human RCC were obtained from previous publications [25,26]. Cluster annotations were also obtained from these publications. For each RCC cluster, a list of marker genes was provided by identifying genes that are over-expressed in the corresponding cluster than all the other clusters. These cluster-specific marker gene sets were used as TAM signatures. In total, eight human TAM signatures were defined (Appendix A). Given an RCC gene expression dataset, the BASE algorithm was used to calculate sample-specific TAM scores for each signature [34,36,39]. The TAM signatures were represented as gene sets without assigning weights to the member genes. In this case, the BASE algorithm degenerated into a method like the single-sample GSEA analysis [40]. A high TAM score indicates that the corresponding TAM cells are strongly infiltrated in the tumor.

### 2.5. LASSO Cox Regression

The TCGA-KIRC dataset was randomly divided into a training and testing set with a 1:1 ratio. The training set was analyzed to identify potential prognostic genes, and both the testing set and the entire set were used for validation. First, univariate Cox-proportional hazards regression analysis was used to evaluate the association between the expression of prognostic genes and overall survival [41]. Genes with a *p* value of <0.05 based on the log-rank test were selected as candidate genes. Second, Least Absolute Shrinkage and Selection Operator (LASSO) Cox regression analysis from the R glmnet package was employed to screen the prognostic genes most associated with overall survival in a multivariate model with 10-fold cross-validation, which resulted in Appendix A. These 27 genes (A2M, ACAT2, AUP1, CCL8, CLDN4, CPM, CRHBP, CXCL1, EIF4EBP1, EREG, FUCA1, GNAS, IFI44, IQGAP2, MEF2A, ORMDL1, PDK4, PINK1, RHOQ, SERPINF1, SLAMF9, SLC20A1, SPAG5, SPINT2, STAT2, TM4SF19, TOR3A) compose the final risk score, which is described as follows:$$\mathrm{Riskscore}=\sum_{i=0}^{n}\beta_{i}x_{i}$$
where $\beta_{i}$ refers to the coefficients of each gene, and $x_{i}$ represents the expression value of the gene.

### 2.6. Survival Analysis

For univariate and multivariate survival analyses, Cox proportional hazards models were calculated using the “coxph” function from the R “survival” package [41]. Survival curves were visualized using Kaplan–Meier curves using the “survfit” function from the R “survival” package [41]. Median immune cell infiltration scores were used to stratify patients into “high” and “low” groups for univariate analyses. For multivariate analyses, an infiltration score of 0 was used as a separator to stratify patients into “high” and “low” groups. Differences in survival distributions in each Kaplan–Meier plot were calculated using a log-rank test using the “survdiff” function from the R “survival” package [41].

### 2.7. Statistical Analyses

The Spearman correlation coefficient (SCC) was reported for all correlation analyses as the assumptions underlying the Pearson correlation (i.e., normal distribution, homoscedasticity, or linearity) were not met. SCC was calculated using the R function cor, and significance was assessed using the cor.test [42]. Principal component analysis (PCA) was performed using the prcomp R function [43]. Principal component coordinates for each sample were extracted using the factoextra R package (https://github.com/kassambara/factoextra, accessed on 30 June 2025). Principal component 1 (PC1) was used to represent TAM infiltration. The sensitivity and specificity of the diagnostic and prognostic prediction models were analyzed by the ROC curve and quantified based on the area under the ROC curve (AUC). All statistical tests were two-sided; *p* values < 0.05 were considered statistically significant. All statistical analyses were performed using R software (version 4.2.0) (https://www.R-project.org/, accessed on 30 June 2025).

## 3. Results

### 3.1. Characterization of Tumor-Associated Macrophage Subtypes and Their Immune Microenvironment in Renal Cell Carcinoma

The TAM signatures in RCC samples exhibited substantial variability, reflecting the molecular heterogeneity within the tumor microenvironment (Figure 2A). Hierarchical clustering analysis revealed distinct expression patterns across samples, suggesting that TAM profiles may contribute to the stratification of tumor subtypes and provide insights into their functional roles. Importantly, the TAM1–TAM8 signatures represent distinct gene sets rather than equal-sized groups of genes or samples. As shown in Appendix A, the number of genes varies across TAM signatures, and no averaging was performed to artificially equalize them. Instead, each TAM subtype was defined by its specific marker genes, and the heatmap illustrates their expression patterns across RCC patient samples.

The relationship among the eight TAM signatures (TAM1, 2, 3, 4, 5, 6, 7, and 8) was evaluated using pairwise Spearman correlation analysis. Strong inter-TAM correlations were observed, except for TAM5 and 6 (Appendix A). To further investigate this distinct behavior, correlation analysis was conducted between TAM signatures and immune cell fractions using the quanTIseq package in R (Figure 2B). This analysis revealed that while most TAM signatures exhibited strong correlations with macrophage populations, TAM5 and 6 displayed lower correlation coefficients with classical macrophage markers and showed closer associations with neutrophils. These findings raised questions regarding their cellular identity and functional role within the tumor microenvironment. TAM1, 2, 3, 4, 7, and 8 have strong positive correlations not only with macrophages but also positively correlated with NK cells and CD8+ T cells, suggesting they may represent TAMs and their synergistic interactions with other immune cells, collectively shaping the immune microenvironment of renal cell carcinoma.

The correlation analysis between the selected TAMs (TAM1, 2, 3, 4, 7, and 8) was conducted, and a strong correlation was shown (Figure 2C). PCA was performed to capture overarching variation among the selected TAMs with PC1 explaining the majority of variance (90.8%), indicating a dominant pattern in TAM distribution (Figure 2D). Correlation analysis between PC1 and selected TAM signatures confirmed that the selected TAMs were strongly associated with PC1 (Figure 2E). Further correlation between PC1 and immune cell populations derived from quanTIseq analysis indicated a strong association between the primary TAM axis and macrophage populations, as well as CD8+ T cells, highlighting the immunomodulatory role of TAMs in RCC (Figure 2F). The identity of these TAM signatures was further validated using an independent deconvolution method, BASE [33], as shown in Appendix A. We also performed the survival analyses for these TAM signatures (Appendix A). The *p* values are not significant in RCC cancers, which are consistent with the previous studies [44,45,46,47]. TAMs play a complex role in cancer progression, with their impact varying across different tumor types. While high TAM density is generally associated with poor prognosis in many cancers, including gastric, breast, bladder, ovarian, oral, and thyroid cancers [46,47], some studies report conflicting results. For instance, in colorectal cancer, high TAM density correlates with better overall survival [46]. The location of TAMs within the tumor microenvironment is crucial, as nest TAMs in gastric cancer are linked to increased tumor cell apoptosis and improved patient survival [44]. The prognostic value of TAMs may also depend on their phenotype (M1 or M2) and the biomarkers used for detection [45]. Overall, the impact of TAMs on patient survival is complex and tumor-specific, highlighting the need for further research to elucidate their role in different cancer types [45].

### 3.2. Principal Component Analysis of Tumor-Associated Macrophage Signatures and Their Immunological Associations in RCC

PC1 showed positive correlations with both M1 and M2 macrophage marker genes, indicating that the selected TAM signatures encompass features of both pro-inflammatory and immunosuppressive macrophage phenotypes (Figure 3A). Additionally, PC1 exhibited strong associations with immune checkpoint gene expression (Figure 3B).

To assess the relationship between TAM-related variance and broader immune responses, PC1 was correlated with key immune-related pathways and scores. Significant correlations were observed with macrophage regulation, leukocyte fraction, stromal fraction, lymphocyte infiltration signature score, IFN-γ response, TCR richness, Transforming growth factor beta (TGF-β) response, and gamma delta T cells (γδ T cells) (Figure 3C–J). These findings suggest that TAM-associated variability is strongly linked to immune infiltration, immune regulation, and broader tumor microenvironment dynamics.

### 3.3. A 27-Gene Risk Score for Prognostic Prediction in RCC

The risk model analysis identified 27 risk-associated genes through LASSO Cox regression, utilizing 10-fold cross-validation to select the lambda value that minimized partial likelihood deviance. These genes exhibited non-zero coefficients, defining a gene set relevant for prognostic prediction (Figure 4A). Kaplan–Meier survival analysis stratified patients into low-risk and high-risk groups based on the risk scores. The survival curves displayed distinct differences in overall survival between the groups, with the high-risk group exhibiting shorter survival durations (Hazard Ratio = 4.64, *p* < 0.001) (Figure 4B). This analysis was performed in the independent validation cohort (E-MTAB-1980), confirming the prognostic robustness of the TAM-risk model across datasets. Time-dependent ROC curves were generated to assess the model’s predictive accuracy for 1-year, 3-year, and 8-year survival, and AUC values of 0.82, 0.79, and 0.82 were observed, respectively, demonstrating consistent predictive performance across time points (Figure 4C). A forest plot illustrated univariate Cox regression results, including the risk score and clinical parameters such as age, gender, tumor stage, and pathologic T stage (Figure 4D). The correlation analysis between the risk scores and immune cell populations, derived from quanTIseq deconvolution, revealed notable patterns (Figure 4E). Higher risk scores were positively associated with M1 macrophages, CD8+ T cells, and Tregs, indicating their enhanced prevalence in patients with elevated risk scores. In contrast, neutrophils and CD4+ T cells showed negative correlations, but these associations were not statistically significant, indicating that immune activation rather than suppression predominantly characterizes the high-risk group’s tumor microenvironment. The relationship between the risk scores and immune pathways was also explored, highlighting key associations with pathways such as Cytotoxic Score, Proliferation, Stromal Fraction, Lymphocyte Infiltration Signature, and Leukocyte Fraction. These pathways showed strong positive correlations with the risk scores, suggesting an enrichment of immune infiltration and stromal activity in patients with higher risk scores (Figure 4F).

### 3.4. The TAM Risk Model Can Evaluate RCC Patients Across Different Clinicopathological Factors

Stratified survival analyses of the risk score model were conducted to evaluate its prognostic value across various clinicopathological factors. In male patients, the Kaplan–Meier survival analysis demonstrated a clear stratification of survival outcomes between the low-risk and high-risk groups, with the high-risk group exhibiting significantly shorter survival times (Figure 5A). Similarly, in female patients, the risk model effectively stratified survival outcomes, further validating its predictive capability across genders (Figure 5B). Age-stratified analyses revealed that the risk model consistently distinguished survival outcomes in both the elderly population (age > 50; Figure 5C) and younger patients (age ≤ 50; Figure 5D). High-risk patients in both age groups displayed poorer survival compared to their low-risk counterparts. The prognostic utility of the risk score model was also examined in relation to tumor stage. Patients with high tumor stages (III and IV) were distinctly stratified into high- and low-risk groups, with significant survival differences (Figure 5E). Similarly, for patients with low tumor stages (I and II), the risk model effectively categorized survival outcomes (Figure 5F). To further dissect the risk model’s predictive ability, the TNM staging system was analyzed. In high T stage (T3–T4) patients, the risk model demonstrated significant stratification of survival outcomes (Figure 5G), while low T stage (T1–T2) patients exhibited comparable results (Figure 5H). For lymph node involvement, the model effectively stratified survival outcomes in patients with low N stage (N0) (Figure 5J). However, stratification was not statistically significant in patients with high N stage (N1–N2), likely due to the small number of patients (n = 16) in this subgroup (Figure 5I). Given this limited sample size, we cannot conclusively determine whether the prognostic significance of the 27-gene risk model truly varies based on lymph node involvement. However, effective survival stratification was observed in patients with low N stage (N0) (Figure 5J). Finally, the risk model demonstrated similar predictive power for distant metastasis, with high M stage (M1) patients showing marked differences in survival outcomes between the risk groups (Figure 5K), while low M stage (M0) patients followed the same trend (Figure 5L). To comprehensively illustrate patient heterogeneity, we analyzed key clinicopathological characteristics within high-risk and low-risk patient groups. Specifically, we provided detailed comparisons, including age, gender distribution, overall pathologic stages (stage I–IV; Figure 5E,F), and detailed TNM categories (pathologic T, N, and M stages). Kaplan–Meier survival analyses (Figure 5) demonstrated significant survival disparities linked with these specific clinical parameters. These results show meaningful associations between the derived risk scores and clinically validated prognostic factors, reinforcing the clinical relevance and predictive robustness of our genomic model.

### 3.5. High-Risk Patients with Significantly Down-Regulated TAMs and Poor Prognosis in RCC

The distribution of risk scores among RCC patients demonstrated a clear stratification between low-risk and high-risk groups (Figure 6A). Patients were ranked by increasing risk scores, with a noticeable separation at the median threshold. Those classified as high-risk exhibited markedly elevated scores compared to their low-risk counterparts. Analysis of survival times revealed distinct differences between patients based on their risk classification (Figure 6B). A scatter plot of survival time against risk scores highlighted a concentration of shorter survival durations among high-risk individuals. Patients classified as “dead” predominantly clustered in the high-risk group, while those still alive were more frequent in the low-risk category. The vertical median line further emphasized the survival disparities between these groups. To explore the relationship between risk scores, PC1 groupings, and survival outcomes, an alluvial diagram was constructed (Figure 6C). This visualization illustrated the transitions between low- and high-risk groups, their associated PC1 classifications, and survival statuses. The diagram demonstrates that both high and low PC1 groups contribute similarly to poor survival outcomes (‘dead’ status). However, the high-risk score category, encompassing patients from both PC1 groups, distinctly shows a greater contribution to unfavorable clinical outcomes, showing the risk score’s stronger association with survival compared to PC1 alone. The difference between high-risk and low-risk patients in their TIME was further examined using the ESTIMATE algorithm, in Immune, Stromal, and ESTIMATE scores, respectively (Figure 6D–F). High-risk patients displayed significantly higher immune scores, indicative of increased immune cell infiltration within the tumor microenvironment (Figure 6D). In contrast, the analysis of Stromal scores did not reveal a statistically significant difference between the high-risk and low-risk groups, suggesting comparable stromal content across both groups (Figure 6E). The ESTIMATE scores, which combine Immune and Stromal scores, were significantly elevated in the high-risk group, reflecting an overall enriched immune and stromal tumor microenvironment in these patients (Figure 6F). These findings align with the notion that high-risk patients exhibit a more active and complex tumor microenvironment.

## 4. Discussion

This study provides a refined characterization of TAMs in RCC and their relationship with TIME. A key finding of this study is the eight TAM signatures, particularly the distinction of TAM5 and 6, which exhibited weaker correlations with other TAM signatures (Appendix A). A number of recent immunoprofiling studies corroborate that not all myeloid cells classified as “TAMs” share lineage markers, implying that certain subsets may indeed be more closely aligned with neutrophils or transitional myeloid phenotypes [48,49]. Further analysis revealed that these subtypes were less associated with macrophage markers and more closely related to neutrophils (Figure 2B), suggesting that they may represent a different myeloid lineage or a transitional phenotype rather than classical TAMs. These data reinforce emerging evidence that tumor-infiltrating myeloid cells are far from uniform, complicating attempts at a simple M1/M2 dichotomy [50,51]. This study indicates the need for a more nuanced categorization when assessing their functional roles in tumor progression.

To avoid confounding effects from these distinct TAMs signatures, we performed PCA using selected TAMs and identified PC1 as a dominant feature explaining the majority of variance (Figure 2D). Interestingly, PC1 correlated with both M1 and M2 macrophage markers (Figure 3A), reinforcing the concept that TAMs in RCC do not conform to a strict M1/M2 dichotomy but rather exist along a spectrum influenced by the tumor microenvironment. Numerous studies have shown that microenvironmental cues, such as cytokines and metabolic byproducts, drive a continuum of macrophage phenotypes with overlapping M1- and M2-like features [52]. Additionally, PC1 showed strong associations with immune checkpoint expression (Figure 3B) and key immune-related scores and pathways (Figure 3C–J), further emphasizing that TAMs play a crucial role in shaping immune regulation within the tumor microenvironment. This aligns with emerging data demonstrating that macrophage-driven checkpoint ligand expression can potentiate T-cell exhaustion and immune evasion in several cancers, including RCC [53,54].

One of the most significant findings of this study is the correlative relationship between TAM activity, immune infiltration, and patient prognosis. The risk model developed in this study stratified patients into high-risk and low-risk groups based on a 27-gene signature, demonstrating a clear survival difference (Figure 4B). Notably, higher risk scores were positively correlated with CD8+ T cells, Tregs, and M1 macrophages (Figure 4E), suggesting that an immune-inflamed tumor environment does not necessarily confer a survival advantage [55,56,57,58]. Recent analyses of T-cell–inflamed tumors show that infiltration alone may be insufficient for clinical benefit, especially where the immunosuppressive axes are dominant [55,56]. This contradicts the conventional assumption that greater immune infiltration predicts better treatment outcomes. Indeed, studies have demonstrated that even large populations of CD8+ T cells or M1-like macrophages can fail to control tumor progression if these populations are functionally exhausted or co-opted by suppressive signals, resulting in a net immunosuppressive microenvironment despite high immune infiltration [56,59]. The strong correlation between high-risk scores and immune-related pathways, including cytotoxic activity and lymphocyte infiltration (Figure 4F), reinforces this notion and highlights the complexity of immune interactions in RCC.

The alignment between risk scores, TAM-associated PC1, and survival outcomes further shows the prognostic significance of TAMs in RCC (Figure 6C). High-risk patients with lower PC1 scores were predominantly associated with poor survival, indicating that TAM activity, rather than overall immune infiltration, is a key determinant of tumor progression. Studies using multiplex immunohistochemistry and single-cell transcriptomics have similarly observed that increased macrophage functional states (e.g., high antigen-presenting but also immunosuppressive signatures) portend worse outcomes [58,59]. Moreover, immune score analysis revealed that high-risk patients exhibited significantly higher immune infiltration (Figure 6D), yet stromal scores did not differ significantly (Figure 6E), suggesting that immune cell composition, rather than stromal content, plays a more decisive role in shaping patient outcomes.

These findings contribute to the growing recognition that TAMs are central players in the immune landscape of RCC. Unlike previous studies that broadly categorized TAMs, our approach provides a more refined classification, emphasizing the need to account for their heterogeneity in both research and clinical applications. Furthermore, our findings suggest that targeting TAMs may be a viable strategy for modulating the immune microenvironment in RCC. Future studies incorporating single-cell RNA sequencing could further delineate TAM plasticity and functional states, while experimental validation could clarify their precise roles in tumor progression. Prospective clinical validation of the risk model in independent cohorts would also strengthen its utility for guiding patient stratification and treatment decisions.

## 5. Conclusions

This study provides a comprehensive characterization of tumor-associated macrophages in RCC, revealing their heterogeneity, immune associations, and prognostic significance. By integrating transcriptomic deconvolution, PCA, and risk modeling, we identified distinct TAM signatures and demonstrated that their activity, rather than overall immune infiltration, plays an important role in shaping patient outcomes. These findings emphasize the need to consider TAM diversity when evaluating immune regulation in RCC and highlight the potential of TAMs as therapeutic targets.

Despite these insights, several limitations should be acknowledged. First, the transcriptomic deconvolution methods used to infer TAM abundance rely on bulk RNA sequencing data, which may obscure intratumoral heterogeneity. Single-cell RNA sequencing or spatial transcriptomics could provide a more granular understanding of TAM subsets and their functional states. Second, while the risk model demonstrated strong prognostic value, further validation in more independent RCC cohorts could be necessary to establish its clinical applicability. Third, although our conclusions are based on computational inference, future studies are warranted to confirm these findings at the protein and cellular levels through immunohistochemical (IHC) validation using RCC tissue slides. Fourth, while scRNA-seq data were used for signature development and validation, matched non-tumor scRNA-seq samples were not available as controls or references for inclusion in this study, which limits comparative interpretation of macrophage signatures in the tumor versus normal context.

Future research should focus on refining TAM classifications through single-cell approaches, exploring functional validation using in vitro and in vivo models, and assessing the therapeutic potential of targeting TAMs to improve RCC treatment strategies. Integrating these findings into personalized medicine approaches may enable more precise risk stratification and immunotherapeutic interventions tailored to TAM-mediated immune dynamics in RCC.

## Figures and Tables

**Figure 1 cells-14-01740-f001:**
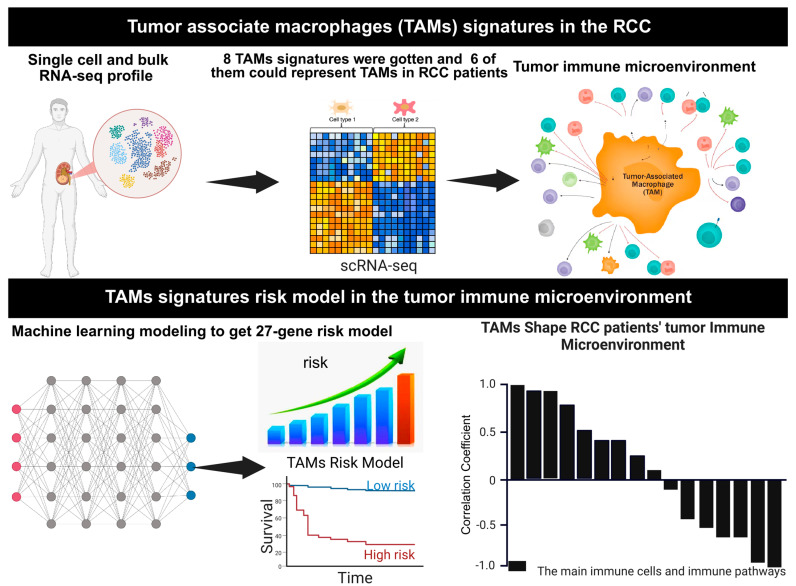
**Study overview.** Utilizing multiple independent single-cell RNA-seq data from human RCC samples, we crafted 8 distinct RCC TAMs signatures, and 6 of them could reflect TAMs infiltration in the RCC. We systematically investigated the role of TAM infiltration in the TIME and the prognosis of the RCC patients. A LASSO Cox regression model was developed for RCC patients’ prognosis and independently validated in multiple RCC patient cohorts. The mechanism of the TAM infiltration affects the prognosis of RCC patients through adjusting the TIME, which was also investigated in this study. Figure was created with BioRender.com.

**Figure 2 cells-14-01740-f002:**
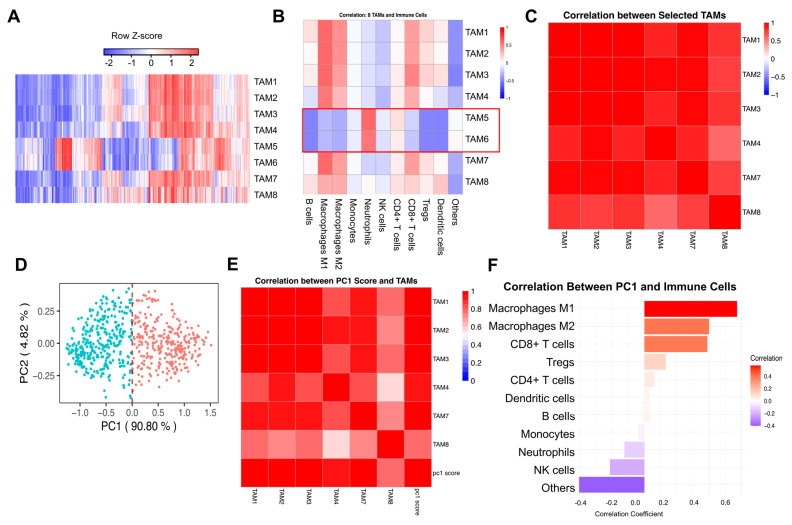
**Tumor-associated macrophage profiles, immune interactions, and principal component analysis in RCC.** (**A**) The heatmap of TAM signature infiltration scores in TCGA-KIRC patients. (**B**) Correlation between TAMs signatures and immune cell fractions. (**C**) Correlation analysis between selected TAM subtypes (TAM1, 2, 3, 4, 7, and 8). (**C**) Correlation between TAM signatures and immune cell fractions. (**D**) PCA of TAM signatures. (**E**) PC1 correlation with TAM signatures. (**F**) PC1 correlation with immune cells. The immune cell fractions in tumor samples were quantified using the quanTIseq method [32].

**Figure 3 cells-14-01740-f003:**
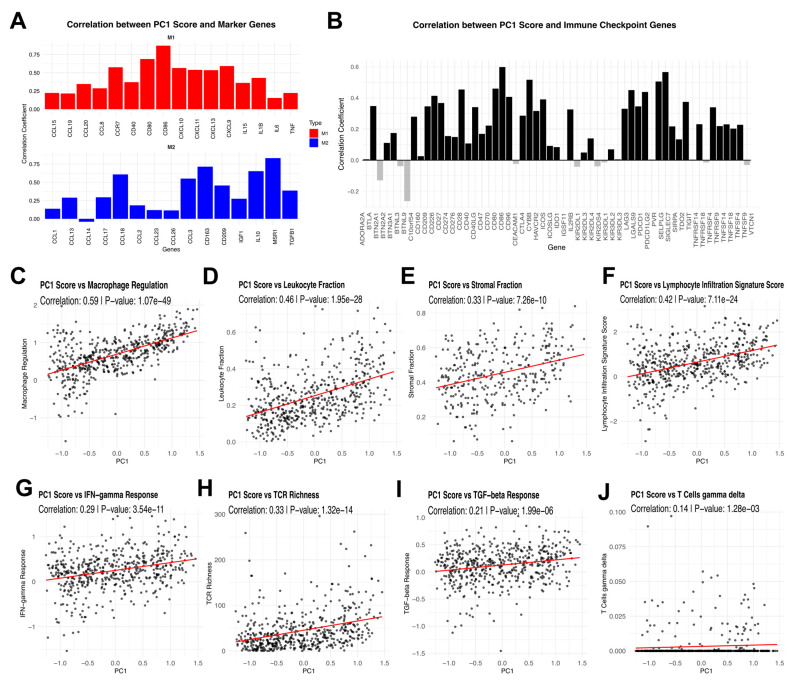
**Principal component analysis of TAM signatures and their immune correlations in RCC.** (**A**) Correlation of PC1 with M1 and M2 marker genes. (**B**) PC1 correlation with immune checkpoint genes. (**C**–**J**) PC1 correlation with immune-related scores and pathways, such as Macrophage Regulation, Leukocyte Fraction, Stromal Fraction, Lymphocyte Infiltration Signature Score, IFN-γ Response, TCR Richness, Transforming growth factor beta (TGF-β) Response, and gamma delta T cells (γδ T cells).

**Figure 4 cells-14-01740-f004:**
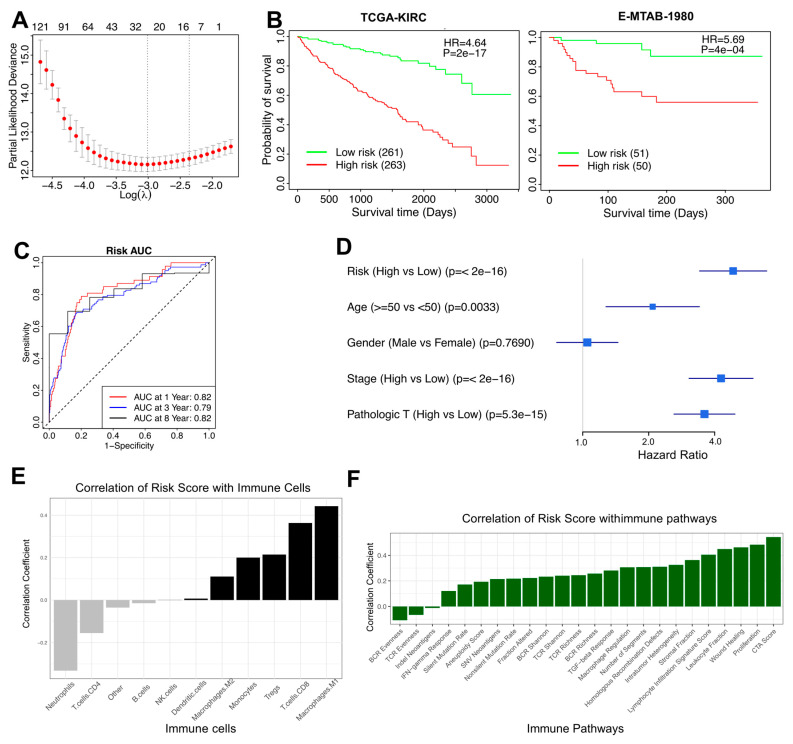
**Risk model analysis for Prognostic Prediction in RCC.** (**A**) Identification of risk genes Using LASSO regression. (**B**) Survival analysis based on risk scores in the TCGA-KIRC patients and the independent validation cohort E-MTAB-1980. (**C**) ROC curves for risk model performance. (**D**) Forest plot of clinical factors and risk scores. (**E**) Risk scores correlation with immune cells. (**F**) Risk scores correlation with immune-related pathways.

**Figure 5 cells-14-01740-f005:**
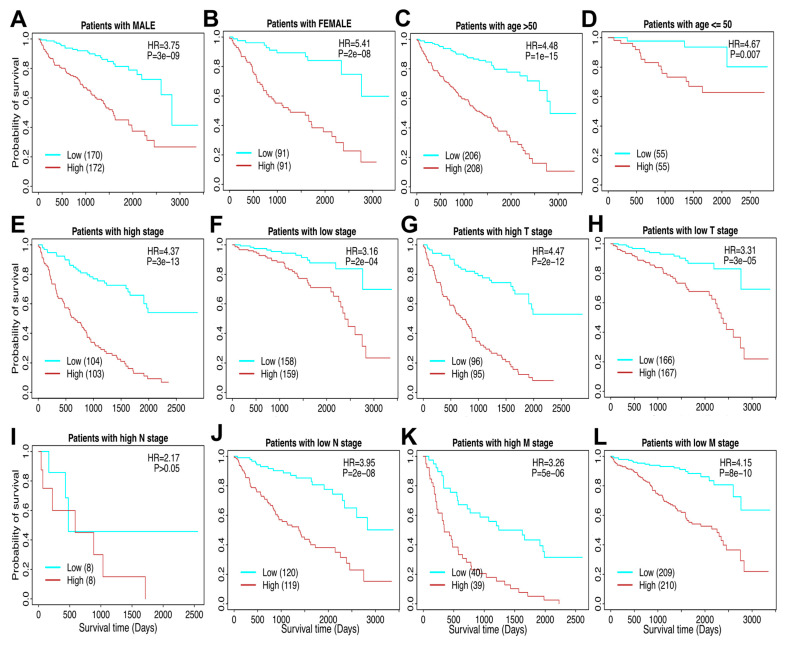
**Stratified survival analysis of the risk score model in clinicopathological factors.** (**A**) The risk model in male patients. (**B**) The risk model in female patients. (**C**) The risk model in the elderly (age > 50). (**D**) The risk model in the young (age ≤ 50). (**E**) The risk model in high tumor stage patients. (**F**) The risk model in low tumor stage patients. For the TNM cancer staging system, TNM stands for Tumor, Nodes, and Metastasis. T is assigned based on the extent of involvement at the primary tumor site, N for the extent of involvement in regional lymph nodes, and M for distant spread. (**G**) The risk model in high T stage patients. (**H**) The risk model in low T stage patients. (**I**) The risk model in high N stage patients. (**J**) The risk model in low N stage patients. (**K**) The risk model in high M stage patients. (**L**) The risk model in low M stage patients.

**Figure 6 cells-14-01740-f006:**
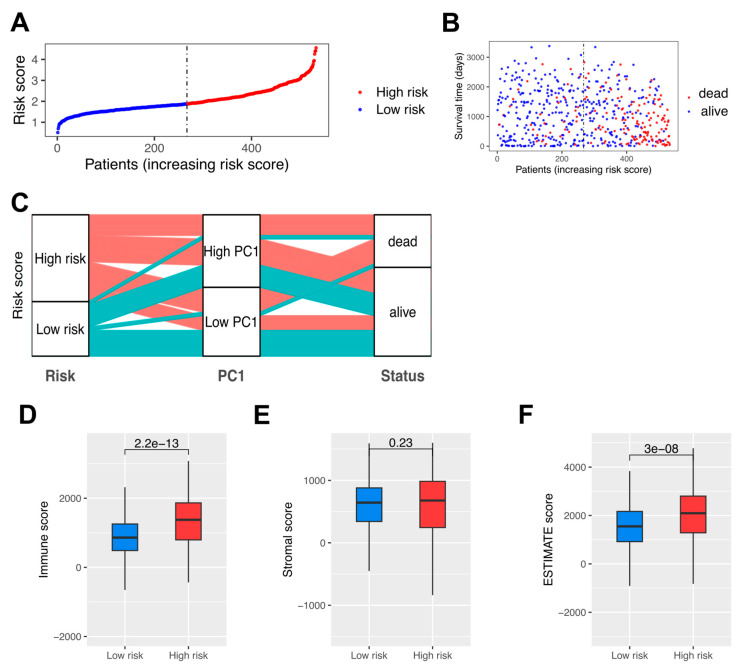
**Risk score stratification and immune profiling of RCC patients.** (**A**) Distribution of risk scores among RCC patients. (**B**) Survival times in low-risk and high-risk patients. (**C**) Alluvial diagram of risk groups, PC1 classifications, and survival outcomes. (**D**) Immune scores in low-risk and high-risk patients. (**E**) Stromal scores in low-risk and high-risk patients. (**F**) ESTIMATE scores in low-risk and high-risk patients.

## Data Availability

All data available in this study is publicly available. The RNA-seq data and clinical information for TCGA-KIRC were obtained from TCGA on FireBrowse (gdac.broadinstitute.org/). All codes in this study and any additional information required in this paper are available from the lead contact upon request.

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
