# Peer review of "Distinct Tumor-Associated Macrophage Signatures Shape the Immune Microenvironment and Patient Prognosis in Renal Cell Carcinoma"

_cells, 2025, doi:10.3390/cells14211740_

Round 1
Reviewer 1 Report
Comments and Suggestions for Authors
1. As noted by the authors, the heterogeneity of RCC is substantial. The authors should provide clinicopathological characteristics of all patients included in the sequencing analysis.
2. This study proposes a bioinformatic model to predict patient prognosis; however, no validation of the model is presented.
3. It is also recommended that the authors perform immunohistochemical (IHC) validation on tissue slides, rather than relying solely on bioinformatic analyses to draw conclusions.
Author Response
#@@@@@@ point to point answer
We are thankful for the reviewer’s overall enthusiasm and constructive comments concerning our manuscript entitled “Distinct Tumor-Associated Macrophage Signatures Shape the Immune Microenvironment and Patient Prognosis in Renal Cell Carcinoma” (ID: cells-3648964). These valuable comments have been helpful in revising and improving our manuscript, as well as providing important guiding significance to this study. We have carefully reviewed the comments and revised the manuscript accordingly, which strengthened the resulting paper considerably. We hope our changes will be met with your approval. The revised sections are highlighted in yellow on the paper.
#@@@ Reviewer 1
#++++++++++++++
- As noted by the authors, the heterogeneity of RCC is substantial. The authors should provide clinicopathological characteristics of all patients included in the sequencing analysis.
#++++++++++++++
Answer: We thank the reviewer for this comment. We added the clinicopathological characteristics of the bulk RNA-seq dataset, TCGA-KIRC into the Supplementary Table 1. The clinicopathological characteristics of the single cell RNA-seq datasets were reported in previous studies. We have added the accompanying description into the Materials and Methods section of the revised manuscript.
We compared the prognosis prediction by using our risk score and the main clinicopathological characteristics, as shown in the Figure 4D. We did survival analyses for the main clinicopathological characteristics of all patients, as shown in the Figure 5. We also investigated the correlation of the risk vs the main clinicopathological characteristics. We found that high-risk patients had poor prognosis, which was independent in all main clinicopathological characteristics, as shown in the Figure 4 and 5. We have added relevant descriptions for them in the results section of the revised manuscript; changes have been highlighted in yellow.
Excerpt From Revised Manuscript:
Section “Materials and Methods”, Line 84, “
Level 3 RNA-seq data and clinical information for The Cancer Genome Atlas Kidney Renal Clear Cell Carcinoma (TCGA-KIRC, tumor samples number = 537) dataset were obtained from TCGA on FireBrowse (gdac.broadinstitute.org/), and their accompanying clinical information is also shown in the Supplementary Table 1. Two independent single-cell RNA-seq datasets and clinical information from human RCC samples were obtained from previous publications [25,26]. The gene expression of 101 RCC patients and related clinical information data, were collected from the Sato et al study named E-MTAB-1980 [27].”
Section “Results”, line 303, “For lymph node involvement, the model effectively stratified survival outcomes in pa-tients with low N stage (N0) (Figure 5J). However, stratification was not statistically sig-nificant in patients with high N stage (N1–N2), likely due to the small number of patients (n = 16) in this subgroup (Figure 5I). Given this limited sample size, we cannot conclu-sively determine whether the prognostic significance of the 27-gene risk model truly varies based on lymph node involvement.”.
Section “Results”, line 312, “To comprehensively illustrate patient heterogeneity, we analyzed key clinicopathological characteristics within high-risk and low-risk patient groups. Specifically, we provided detailed comparisons, including age, gender distribution, overall pathologic stages (stage I–IV; Figures 5E and 5F), and detailed TNM categories (pathologic T, N, and M stages). Kaplan-Meier survival analyses (Figure 5) demonstrated significant survival disparities linked with these specific clinical parameters. These results show meaningful associations between the derived risk scores and clinically validated prognostic factors, reinforcing the clinical relevance and predictive robustness of our genomic model.”
#++++++++++++++
- This study proposes a bioinformatic model to predict patient prognosis; however, no validation of the model is presented.
#++++++++++++++
Answer: We thank the reviewer for the kind suggestion. We added an independent dataset and validated our risk model in the independent RCC patient datasets, which enhanced the persuasiveness of the model. We further observed that high-risk patients had worse prognosis than low-risk patients.
We also updated the results of the comparison analyses between low-risk and high-risk patients. Comparison analyses further revealed that the low-risk patients had significantly up-regulated immune-related pathways in RCC patients, as shown in the Figure 4B of the revised manuscript. These details are now explicitly described in the results section of the revised manuscript.
Excerpt From Revised Manuscript:
Section “Materials and Methods”, line 89, “The gene expression of 101 RCC patients and related clinical information data, were collected from the Sato et al study named E-MTAB-1980 [27].”
Section “Results”, line 260, “This analysis was performed in the independent validation cohort (E-MTAB-1980), confirming the prognostic robustness of the TAMs-risk model across datasets.”
Figure 4. Risk Model Analysis for Prognostic Prediction in RCC. (A) Identification of Risk Genes Using LASSO Regression. (B) Survival Analysis Based on Risk Scores in the TCGA-KIRC Patients and the independent validation cohort E-MTAB-1980. (C) ROC Curves for Risk Model Performance. (D) Forest Plot of Clinical Factors and Risk Scores. (E) Risk Scores Correlation with Immune Cells. (F) Risk Scores Correlation with Immune Related Pathways.
#++++++++++++++
- It is also recommended that the authors perform immunohistochemical (IHC) validation on tissue slides, rather than relying solely on bioinformatic analyses to draw conclusions.
#++++++++++++++
Answer: We appreciate this important suggestion. We agree that immunohistochemical (IHC) on tissue sections would provide orthogonal, spatially resolved validation of our findings. At present, we do not have access to sufficient archived tissue or prospective tissue collection under our current IRB to perform IHC. We have added the validation into the limitation of the discussion section of the revised manuscript, which have been highlighted in yellow.
Excerpt From Revised Manuscript:
Section Discussion, Line 444, “Third, although our conclusions are based on computational inference, future studies are warranted to confirm these findings at the protein and cellular levels through im-munohistochemical (IHC) validation using RCC tissue slides.”

Reviewer 2 Report
Comments and Suggestions for Authors
(1) The authors used scRNA-seq data from RCC patient samples and performed hierarchical clustering analysis to categorize TAMs into eight subtypes. Subsequently, they employed LASSO Cox regression analysis to identify a 27-gene signature that was used to construct a risk prediction model for poor overall survival in RCC patients. The authors should provide a clear flowchart or schematic diagram summarizing the filtering and selection process leading from the eight TAM subtypes to the final 27 genes, which would greatly enhance clarity and reproducibility.
(2) According to Figure 5, the prognostic significance of the 27-gene risk model appears to be dependent on N-stage, while it remains independent of gender, age, T-stage, and M-stage. Although no experimental validation was conducted, the authors are encouraged to include a discussion of these findings in the manuscript, particularly regarding the possible biological implications and clinical relevance of N-stage-dependent effects.
(3) In the Materials and Methods section, the authors should first state how many RCC patient samples and how many scRNA-seq data they used for the study. Additionally, it should be clarified whether non-tumor samples were included as controls or references for the analysis.
(4) Several figures contain labels that are too small and even have low resolution, which affects readability. The authors should improve the figure quality to ensure clarity and legibility, particularly for labels, legends, and axis texts.
Author Response
#@@@@@@ point to point answer
We are thankful for the reviewer’s overall enthusiasm and constructive comments concerning our manuscript entitled “Distinct Tumor-Associated Macrophage Signatures Shape the Immune Microenvironment and Patient Prognosis in Renal Cell Carcinoma” (ID: cells-3648964). These valuable comments have been helpful in revising and improving our manuscript, as well as providing important guiding significance to this study. We have carefully reviewed the comments and revised the manuscript accordingly, which strengthened the resulting paper considerably. We hope our changes will be met with your approval. The revised sections are highlighted in yellow on the paper.
#@@@ Reviewer 2
The authors used scRNA-seq data from RCC patient samples and performed hierarchical clustering analysis to categorize TAMs into eight subtypes. Subsequently, they employed LASSO Cox regression analysis to identify a 27-gene signature that was used to construct a risk prediction model for poor overall survival in RCC patients.
#++++++++++++++
- The authors should provide a clear flowchart or schematic diagram summarizing the filtering and selection process leading from the eight TAM subtypes to the final 27 genes, which would greatly enhance clarity and reproducibility.
#++++++++++++++
Answer: We thank the reviewer’s kind suggestion. We agree that a schematic diagram would improve clarity. We modified Figure 1 and added the filtering and selection process into the Figure 1 of the revised manuscript, as the reviewer suggested.
Excerpt From Revised Manuscript:
The modified Figure 1 and figure legends are shown below.
Figure 1. Study overview. Utilizing multiple independent single-cell RNA-seq data from human RCC samples, we crafted 8 distinct RCC TAMs signatures, and 6 of them could reflect TAMs infiltration in the RCC. We systematically investigated the role of TAMs infiltration in the TIME and the prognosis of the RCC patients. A LASSO Cox regression model was developed for RCC patients' prognosis, and independently validated in multiple RCC patient cohorts. The mechanism of the TAMs infiltration affects the prognosis of RCC patients through adjusting the TIME, which was also investigated in this study.
Section “Introduction”, line 63, “This study focuses on elucidating the roles of TAMs signatures in RCC, leveraging their expression profiles to stratify patients based on risk scores, and investigating their in-teractions with immune components within the TME. By integrating computational de-convolution methods and survival analyses, this work aims to provide insights into the prognostic and therapeutic significance of TAMs in RCC. These findings could pave the way for the development of TAM-targeted therapies, offering new avenues for improving outcomes in RCC patients. A 27-gene risk score derived from TAMs signatures across independent datasets can effectively classify patients into low- and high-risk categories, distinguish overall survival, and investigate the potential mechanisms underlying the role of TAMs in the prognosis of RCC patients. The workflow of the present study is il-lustrated in Figure 1.”
#++++++++++++++
- According to Figure 5, the prognostic significance of the 27-gene risk model appears to be dependent on N-stage, while it remains independent of gender, age, T-stage, and M-stage. Although no experimental validation was conducted, the authors are encouraged to include a discussion of these findings in the manuscript, particularly regarding the possible biological implications and clinical relevance of N-stage-dependent effects.
#++++++++++++++
Answer: We thank the reviewer’s kind suggestion. We expanded on the observation that the prognostic significance of the 27-gene risk model is dependent on N-stage. We discussed the potential biological implications of this finding and the N-stage-dependent effects relevance to clinical practice, particularly in predicting lymph node involvement and metastasis.
Excerpt From Revised Manuscript:
Section “Results”, Line 303, “For lymph node involvement, the model effectively stratified survival outcomes in pa-tients with low N stage (N0) (Figure 5J). However, stratification was not statistically sig-nificant in patients with high N stage (N1–N2), likely due to the small number of patients (n = 16) in this subgroup (Figure 5I). Given this limited sample size, we cannot conclu-sively determine whether the prognostic significance of the 27-gene risk model truly varies based on lymph node involvement.”
#++++++++++++++
- In the Materials and Methods section, the authors should first state how many RCC patient samples and how many scRNA-seq data they used for the study. Additionally, it should be clarified whether non-tumor samples were included as controls or references for the analysis.
#++++++++++++++
Answer: We thank the reviewer’s kind comment. We stated the RCC samples information of the study into the Materials and Methods section, as the reviewer suggested. We did not include the non-tumor samples as controls or references in this study. We have modified the according description, as shown in the revised manuscript. Thank you.
Excerpt From Revised Manuscript:
Section “Materials and Methods”, Line 84, “Level 3 RNA-seq data and clinical information for The Cancer Genome Atlas Kidney Renal Clear Cell Carcinoma (TCGA-KIRC, tumor samples number = 537) dataset were obtained from TCGA on FireBrowse (gdac.broadinstitute.org/), and their accompanying clinical information is also shown in the Supplementary Table 1. Two independent sin-gle-cell RNA-seq datasets and clinical information from human RCC samples were ob-tained from previous publications [25,26]. The gene expression of 101 RCC patients and related clinical information data, were collected from the Sato et al study named E-MTAB-1980 [27].”
Section “Conclusions”, Line 447, “Fourth, while scRNA-seq data were used for signature development and validation, matched non-tumor scRNA-seq samples were not available as controls or references for inclusion in this study, which limits comparative interpretation of macrophage signatures in the tumor versus normal context.”
#++++++++++++++
- Several figures contain labels that are too small and even have low resolution, which affects readability. The authors should improve the figure quality to ensure clarity and legibility, particularly for labels, legends, and axis texts.
#++++++++++++++
Answer: We thank the reviewer’s kind comments. We improved the figures quality of the Figure 1, Figure 2, Figure 3A, Figure 4, Figure 6, and Supplemental Figure 1 to increase clarity and legibility. The updated figures are shown in the revised manuscript.
Excerpt From Revised Manuscript:
Here are the quality improved figures, as shown follows.
Figure 1. Study overview. Utilizing multiple independent single-cell RNA-seq data from human RCC samples, we crafted 8 distinct RCC TAMs signatures, and 6 of them could reflect TAMs infiltration in the RCC. We systematically investigated the role of TAMs infiltration in the TIME and the prognosis of the RCC patients. A LASSO Cox regression model was developed for RCC patients' prognosis, and independently validated in multiple RCC patient cohorts. The mechanism of the TAMs infiltration affects the prognosis of RCC patients through adjusting the TIME, which was also investigated in this study. Figure created with BioRender.com.
Figure 2. Tumor-Associated Macrophage Profiles, Immune Interactions, and Principal Component Analysis in RCC. (A) The heatmap of TAMs signatures infiltration scores in TCGA-KIRC patients. (B) Correlation Between TAMs Signatures and Immune Cell Fractions. (C) Correlation Analysis Between Selected TAMs Subtypes (TAM1, 2, 3, 4, 7, and 8). (C) Correlation Between TAMs Signatures and Immune Cell Fractions. (D) PCA of TAMs Signatures. (E) PC1 Correlation with TAMs Signatures. (F) PC1 Correlation with Immune Cells. The immune cell fractions in tumor samples were quantified using the quanTIseq method [22].
Figure 3. Principal Component Analysis of TAMs Signatures and Their Immune Correlations in RCC. (A) Correlation of PC1 with M1 and M2 Marker Genes. (B) PC1 Correlation with Immune Checkpoint Genes. (C - J) PC1 Correlation with Immune Related Scores and Pathways, such as Macrophage Regulation, Leukocyte Fraction, Stromal Fraction, Lymphocyte Infiltration Signature Score, IFN-γ Response, TCR Richness, Transforming growth factor beta (TGF-β) Response, and gamma delta T cells (γδ T cells).
Figure 4. Risk Model Analysis for Prognostic Prediction in RCC. (A) Identification of Risk Genes Using LASSO Regression. (B) Survival Analysis Based on Risk Scores in the TCGA-KIRC Patients and the independent validation cohort E-MTAB-1980. (C) ROC Curves for Risk Model Performance. (D) Forest Plot of Clinical Factors and Risk Scores. (E) Risk Scores Correlation with Immune Cells. (F) Risk Scores Correlation with Immune Related Pathways.
Figure 6. Risk Score Stratification and Immune Profiling of RCC Patients. (A) Distribution of risk scores among RCC patients. (B) Survival times in low-risk and high-risk patients. (C) Alluvial diagram of risk groups, PC1 classifications, and survival outcomes. (D) Immune scores in low-risk and high-risk patients. (E) Stromal scores in low-risk and high-risk patients. (F) ESTIMATE scores in low-risk and high-risk patients.
SI Figure. The Independent method (BASE) validate the TAM signatures with important immune cells. (A) Correlation Between TAM Signatures and KIRC Immune Cells: This heatmap demonstrates the correlation between the 8 TAM signatures and six major immune cell types from the KIRC dataset. Except TAM 5 and 6, positive associations are observed, particularly with macrophages, NK cells, and CD8+ T cells, suggesting TAM involvement in RCC immune regulation. (B) PC1 Correlation with Immune Cells: Correlation analysis validated the association between PC1 score and immune cell populations derived from independent deconvolution methods (KIRC immune cells). PC1 showed strong correlations with macrophages, CD8+ T cells, and NK cells, supporting the interplay between the selected TAMs and these key immune effector cells. (C) Survival plot of the TAM1,2,3,4,7,8 signatures. (D) Survival plot of the PC1.

Reviewer 3 Report
Comments and Suggestions for Authors
This manuscript aims to investigate the role of tumor-associated macrophage in renal cell carcinoma. Several analyses is solid, but the technical description of analyses should be better.
- While authors are characterizing TAMs in the paper, they do not capture the difference between TAMs and Ms (non-tumor-associated macrophages). This seems to undermine the use of scRNA-seq, where a large number of factors are not tumor-specific.
- There are several technical references missing, like TCGA, R, and “survival” package.
- Section 2.1: Authors should report if there is any filtering, and the total number of samples after filtering needs to be shown for reproducibility.
- Section 3.1: Authors should elaborate the identification of TAM subtypes seem. It is not clear if authors are subtyping genes or samples even. Figure 2A suggests that the numbers of elements in TAM1-8 are equal, and if any averaging was performed, it should be clearly mentioned.
- Section 3.1: What is the need to go for 8 subtypes in this section? It looks like there are only two types, macrophage and neutrophils.
- Lines 227-229 “Additionally, PC1 exhibited strong associations with immune checkpoint gene expression, suggesting a link between TAM abundance and regulatory immune signaling mechanisms”: There is no data about TAM abundance in this section, and this is a clear overstatement, which needs to be removed.
- Section 3.3: Authors should show how to quantify the risk score as shown in Fig 6A. The weight of different genes among 27 will provide insights on what is the key underlying processes. Also, the list of genes should be provided.
Author Response
#@@@@@@ point to point answer
We are thankful for the reviewer’s overall enthusiasm and constructive comments concerning our manuscript entitled “Distinct Tumor-Associated Macrophage Signatures Shape the Immune Microenvironment and Patient Prognosis in Renal Cell Carcinoma” (ID: cells-3648964). These valuable comments have been helpful in revising and improving our manuscript, as well as providing important guiding significance to this study. We have carefully reviewed the comments and revised the manuscript accordingly, which strengthened the resulting paper considerably. We hope our changes will be met with your approval. The revised sections are highlighted in yellow on the paper.
#@@@ Reviewer 3
This manuscript aims to investigate the role of tumor-associated macrophage in renal cell carcinoma. Several analyses is solid, but the technical description of analyses should be better.
#++++++++++++++
- While authors are characterizing TAMs in the paper, they do not capture the difference between TAMs and Ms (non-tumor-associated macrophages). This seems to undermine the use of scRNA-seq, where a large number of factors are not tumor-specific.
#++++++++++++++
Answer: We thank the reviewer’s insightful comment. We agree that there are many factors are not tumor specific. We acknowledge there are difference between TAMs and non-tumor-associated macrophages (Ms). In this study, we did not include the non-tumor samples as controls or references. We focused on characterizing TAMs within the tumor microenvironment based on their unique gene expression profiles and functional roles in the RCC. We have modified the according description and add it into the limitation parts, as shown in the revised manuscript. Thank you.
Excerpt From Revised Manuscript:
Section “Conclusions”, Line 447, “Fourth, while scRNA-seq data were used for signature development and validation, matched non-tumor scRNA-seq samples were not available as controls or references for inclusion in this study, which limits comparative interpretation of macrophage signatures in the tumor versus normal context.”
#++++++++++++++
- There are several technical references missing, like TCGA, R, and “survival” package.
#++++++++++++++
Answer: We thank the reviewer for the comments. In the revised manuscript, we have added the technical references and links for the missing software and packages. The revised parts have been marked with yellow color.
Excerpt From Revised Manuscript:
Line 155, “For univariate and multivariate survival analyses, Cox proportional hazards models were calculated using the “coxph” function from the R “survival” package [43]. Survival curves were visualized using Kaplan-Meier curves using the “survfit” function from the R “survival” package [43]. Median immune cell infiltration scores were used to stratify patients into “high” and “low” groups for univariate analyses. For multivariate analyses, an infiltration score of 0 was used as a separator to stratify patients into “high” and “low” groups. Differences in survival distributions in each Kaplan-Meier plot were calculated using a log-rank test using the “survdiff” function from the R “survival” package [43].”
Line 164, “The Spearman correlation coefficient (SCC) was reported for all correlation analyses as the assumptions underlying the Pearson correlation (i.e., normal distribution, homo-scedasticity, or linearity) were not met. SCC was calculated using the R function cor, and significance was assessed using the cor.test [44]. Principal component analysis (PCA) was performed using the prcomp R function[45]. Principal component coordinates for each sample were extracted using the factoextra R package (https://github.com/kassambara/factoextra). Principal component 1 (PC1) was used to represent TAMs infiltration. The sensitivity and specificity of the diagnostic and prog-nostic prediction models were analyzed by the ROC curve and quantified based on the area under the ROC curve (AUC). All statistical tests were two-sided; P values < 0.05 were considered statistically significant. All statistical analyses were performed using R soft-ware (version 4.2.0)(https://www.R-project.org/.).”
#++++++++++++++
- Section 2.1: Authors should report if there is any filtering, and the total number of samples after filtering needs to be shown for reproducibility.
#++++++++++++++
Answer: We thank the reviewer for identifying this oversight, we have added more description about the data source, total number samples of the bulk RNAseq and scRNAseq data. They were shown in the “Materials and Methods” section 2.1. Data utilization part of the revised manuscript (line 83), as shown follows.
Excerpt From Revised Manuscript:
Section “Materials and Methods”, line 83, “Level 3 RNA-seq data and clinical information for The Cancer Genome Atlas Kidney Renal Clear Cell Carcinoma (TCGA-KIRC, tumor samples number = 537) dataset were obtained from TCGA on FireBrowse (gdac.broadinstitute.org/), and their accompanying clinical information is also shown in the Supplementary Table 1. Two independent sin-gle-cell RNA-seq datasets and clinical information from human RCC samples were ob-tained from previous publications [25,26]. The gene expression of 101 RCC patients and related clinical information data, were collected from the Sato et al study named E-MTAB-1980 [27]. Macrophage regulation scores, leukocyte and lymphocyte infiltration scores, and IFN-γ response and TGFβ response scores for TCGA-KIRC samples were downloaded as a supplementary file (Supplementary Table 2) from prior work [28].”
#++++++++++++++
- Section 3.1: Authors should elaborate the identification of TAM subtypes seem. It is not clear if authors are subtyping genes or samples even. Figure 2A suggests that the numbers of elements in TAM1-8 are equal, and if any averaging was performed, it should be clearly mentioned.
#++++++++++++++
Answer: We thank the reviewer for their suggestions. The TAM1–TAM8 signatures represent distinct gene sets rather than equal-sized groups of genes or samples. As shown in Supplementary Table 3, the number of genes varies across TAM signatures, and no averaging was performed to artificially equalize them. According to the reviewer’s suggestion, we have added more description about the Figure 2A and we also improved the quality of the Figure 2, in the revised manuscript.
Excerpt From Revised Manuscript:
Section “Results” 3.1, line 181, “Importantly, the TAM1–TAM8 signatures represent distinct gene sets rather than equal-sized groups of genes or samples. As shown in Supplementary Table 3, the number of genes varies across TAMs signatures, and no averaging was performed to artificially equalize them. Instead, each TAMs subtype was defined by its specific marker genes, and the heatmap illustrates their expression patterns across RCC patient samples.”
Figure 2. Tumor-Associated Macrophage Profiles, Immune Interactions, and Principal Component Analysis in RCC. (A) The heatmap of TAMs signatures infiltration scores in TCGA-KIRC patients. (B) Correlation Between TAMs Signatures and Immune Cell Fractions. (C) Correlation Analysis Between Selected TAMs Subtypes (TAM1, 2, 3, 4, 7, and 8). (C) Correlation Between TAMs Signatures and Immune Cell Fractions. (D) PCA of TAMs Signatures. (E) PC1 Correlation with TAMs Signatures. (F) PC1 Correlation with Immune Cells. The immune cell fractions in tumor samples were quantified using the quanTIseq method [22].
#++++++++++++++
- Section 3.1: What is the need to go for 8 subtypes in this section? It looks like there are only two types, macrophage and neutrophils.
#++++++++++++++
Answer: We thank the reviewer’s kind comment. To avoid overstatement regarding the TAM signatures we identified, we retain these gene-expression-based macrophage subpopulations. We have modified Figure 1 and the description of the TAM signatures selection process. They are shown in the Figure 1 and the “Results” section 3.1 of the revised manuscript, as shown follows.
Excerpt From Revised Manuscript:
Section “Results”, line 181, “Importantly, the TAM1–TAM8 signatures represent distinct gene sets rather than equal-sized groups of genes or samples. As shown in Supplementary Table 3, the number of genes varies across TAMs signatures, and no averaging was performed to artificially equalize them. Instead, each TAMs subtype was defined by its specific marker genes, and the heatmap illustrates their expression patterns across RCC patient samples.
The relationship among the eight TAMs signatures (TAM1, 2, 3, 4, 5, 6, 7, and 8) was evaluated using pairwise Spearman correlation analysis. Strong inter-TAMs correlations were observed, except for TAM5 and 6 (Supplemental Figure 1A). To further investigate this distinct behavior, correlation analysis was conducted between TAMs signatures and immune cell fractions using the quanTIseq package in R (Figure 2B). This analysis re-vealed that while most TAMs signatures exhibited strong correlations with macrophage populations, TAM5 and 6 displayed lower correlation coefficients with classical macro-phage markers and showed closer associations with neutrophils. These findings raised questions regarding their cellular identity and functional role within the tumor micro-environment. TAM1, 2, 3, 4, 7, and 8 have strong positive correlations not only with macrophages, but also positively correlated with NK cells and CD8+ T cells, suggesting they may represent TAMs and their synergistic interactions with other immune cells, collectively shaping the immune microenvironment of renal cell carcinoma.
The correlation analysis between the selected TAMs (TAM1, 2, 3, 4, 7, and 8) was conducted, and a strong correlation was shown (Figure 2C).”
#++++++++++++++
- Lines 227-229 “Additionally, PC1 exhibited strong associations with immune checkpoint gene expression, suggesting a link between TAM abundance and regulatory immune signaling mechanisms”: There is no data about TAM abundance in this section, and this is a clear overstatement, which needs to be removed.
#++++++++++++++
Answer: We thank the reviewer’s kind comment. To avoid the overstatement about the TAM abundance in this section, we deleted the “TAM abundance” related part in the manuscript and the description about the Figure 3B. They were shown in the section “Results” 3.2 of the revised manuscript, as shown follows. Thank you.
Excerpt From Revised Manuscript:
Section “Results” 3.2, line 233, “PC1 showed positive correlations with both M1 and M2 macrophage marker genes, indicating that the selected TAMs signatures encompass features of both pro-inflammatory and immunosuppressive macrophage phenotypes (Figure 3A). Ad-ditionally, PC1 exhibited strong associations with immune checkpoint gene expression (Figure 3B).”
#++++++++++++++
- Section 3.3: Authors should show how to quantify the risk score as shown in Fig 6A. The weight of different genes among 27 will provide insights on what is the key underlying processes. Also, the list of genes should be provided.
#++++++++++++++
Answer: We thank the reviewer’s kind suggestion. We added more details description about how to quantify the risk score for a patient of the Fig 6A, as shown in the Methods section. We save the weight of the 27 genes and genes names into the Supplementary Table 3. These 27 genes were identified by the Lasso Cox-regression analysis, and were composed the final risk score for each patient. The modification was shown in the revised manuscript. Thank you.
Excerpt From Revised Manuscript:
Section “Materials and Methods” 2.5, line 140, “Genes with a P value of < 0.05 based on the log-rank test were selected as candidate genes. Second, Least Absolute Shrinkage and Selection Operator (Lasso) Cox-regression analysis from the R glmnet package was employed to screen the prognostic genes most associated with overall survival in a multivariate model with 10-fold cross-validation, which resulted in Supplementary Table 4. These 27 genes, which include: (A2M, ACAT2, AUP1, CCL8, CLDN4, CPM, CRHBP, CXCL1, EIF4EBP1, EREG, FUCA1, GNAS, IFI44, IQGAP2, MEF2A, ORMDL1, PDK4, PINK1, RHOQ, SERPINF1, SLAMF9, SLC20A1, SPAG5, SPINT2, STAT2, TM4SF19, TOR3A), composed the final risk score, which is described as follows:

where refers to the coefficients of each gene and represents the expression value of the gene.
The added references:
- Coulton, A.; Murai, J.; Qian, D.; Thakkar, K.; Lewis, C.E.; Litchfield, K. Using a Pan-Cancer Atlas to Investigate Tumour Associated Macrophages as Regulators of Immunotherapy Response. Nat Commun 2024, 15, 5665, doi:10.1038/s41467-024-49885-8.
- Govindarajan, A.; ... Characterization of Papillary and Clear Cell Renal Cell Carcinoma through Imaging Mass Cytometry Reveals Distinct Immunologic Profiles. Front Immunol 2023, 14, 1182581, doi:10.3389/fimmu.2023.1182581.
- Wei, C.; ... Tumor-Associated Macrophage Clusters Linked to Immunotherapy in a Pan-Cancer Census. NPJ Precis Oncol 2024, 8, 176, doi:10.1038/s41698-024-00660-4.
- Alchahin, A.M.; ... A Transcriptional Metastatic Signature Predicts Survival in Clear Cell Renal Cell Carcinoma. Nat Commun 2022, 13, 5747, doi:10.1038/s41467-022-33375-w.
- Zhang, D.; ... Spatial Heterogeneity of Tumor Microenvironment Influences the Prognosis of Clear Cell Renal Cell Carcinoma. J Transl Med 2023, 21, 489, doi:10.1186/s12967-023-04336-8.
- Liu, H.; ... Molecular Understanding and Clinical Aspects of Tumor-Associated Macrophages in the Immunotherapy of Renal Cell Carcinoma. Journal of Experimental & Clinical Cancer Research 2024, 43, 286, doi:10.1186/s13046-024-03164-y.
- Raghubar, A.M.; ... High-Risk Clear Cell Renal Cell Carcinoma Microenvironments Are Enriched for pro-Tumor Immune Phenotypes. NPJ Precis Oncol 2023, 7, 53, doi:10.1038/s41698-023-00441-5.
- Kourtis, N.; ... A Single-Cell Map of Dynamic Chromatin Landscapes of Immune Cells in Renal Cell Carcinoma. Nat Cancer 2022, 3, 1164–1180, doi:10.1038/s43018-022-00391-0.
- Xie, Y.; ... High CD204+ Tumor-Associated Macrophage Density Predicts a Poor Prognosis in Patients with Clear Cell Renal Cell Carcinoma. J Cancer 2024, 15, 1511–1522, doi:10.7150/jca.91928.
- Chakiryan, N.H.; ... Geospatial Characterization of Immune Cell Distributions and Dynamics across the Microenvironment in Clear Cell Renal Cell Carcinoma. J Immunother Cancer 2023, 11, e006195, doi:10.1136/jitc-2022-006195.
- Shen, A.; Garrett, A.; Chao, C.; Liu, D.; Zhu, Y.; Mai, J.; Jiang, C. A Comprehensive Meta-Analysis of Tissue Resident Memory T Cell Shaping Non-Small-Cell Lung Cancer Immune Microenvironment and Patient Prognosis. Journal of Clinical Oncology 2024, 42, 210–210, doi:10.1200/JCO.2024.42.23_suppl.210.
- Jiang, C.; Li, J.; Helman, E.; Shen, X.; Cheng, C. Abstract 3119: A Novel Tissue Resident Memory T Cell (TRM) Signature Predicts Prognosis and Tumor Microenvironment of Patients with Melanoma. Cancer Res 2023, 83, 3119–3119, doi:10.1158/1538-7445.AM2023-3119.
- Jolliffe, I.T.; Cadima, J. Principal Component Analysis: A Review and Recent Developments. Philosophical Transactions of the Royal Society A: Mathematical, Physical and Engineering Sciences 2016, 374, 20150202, doi:10.1098/rsta.2015.0202.
- Lee Rodgers, J.; Nicewander, W.A. Thirteen Ways to Look at the Correlation Coefficient. Am Stat 1988, 42, 59–66, doi:10.1080/00031305.1988.10475524.
- Fox, J.; Weisberg, S. Cox Proportional-Hazards Regression for Survival Data in R;

Round 2
Reviewer 1 Report
Comments and Suggestions for Authors
The authors have addressed all the questions and made the corresponding revisions. Publication is recommended.
Reviewer 2 Report
Comments and Suggestions for Authors
The manuscript has been revised and improved for publication.